# Partial Substitution of Fish Oil with Microalgae (*Schizochytrium* sp.) Can Improve Growth Performance, Nonspecific Immunity and Disease Resistance in Rainbow Trout, *Oncorhynchus mykiss*

**DOI:** 10.3390/ani12091220

**Published:** 2022-05-09

**Authors:** Seunghan Lee, Cheol-Oh Park, Wonsuk Choi, Jinho Bae, Jiyoung Kim, Sera Choi, Kumar Katya, Kang-Woong Kim, Sungchul C. Bai

**Affiliations:** 1Aquafeed Research Center, National Institute of Fisheries Science, Pohang 37517, Korea; shlee5863@naver.com (S.L.); kangwoongkim@korea.kr (K.-W.K.); 2Feeds & Foods Nutrition Research Center (FFNRC), Pukyong National University, Busan 48513, Korea; 704cohpark@daum.net (C.-O.P.); thm622@naver.com (W.C.); bjh2921@naver.com (J.B.); 3CJ CheilJedang R&D, 55, Gwanggyo-ro 42beon-gil, Suwon-si 16495, Korea; jy.kim2@cj.net; 4CJ CheilJedang BIO, 73, Yangcheon-ro, Seoul 04560, Korea; sera.choi@cj.net; 5Malaysian Aquaponics Research Centre Sdn. Bhd, Kuala Lumpur 50100, Malaysia; kumarkatya85@gmail.com

**Keywords:** sustainability, algae oil, fatty acids, trout feed, digestibility, growth

## Abstract

**Simple Summary:**

The global price of fish oil continued to increase due to a consistent downward production trend. Therefore, research on the replacement of fish oil is growing and the search for sustainable alternative sources in aquaculture. The present study was conducted to evaluate dietary microalgae, *Schizochytrium* sp. (SC) as fish oil replacer in rainbow trout, *Oncorhynchus mykiss*. Our results suggested that dietary SC is well-digested and could replace up to 80% of fish oil in the diet of rainbow trout without negative effects on growth and immune responses.

**Abstract:**

The price of fish oil has reached a historical peak due to a consistent downward production trend, and therefore, the search for sustainable alternative sources has received great attention. This research was conducted to evaluate dietary micro-algae, *Schizochytrium* sp. (SC) as fish oil (FO) replacer in rainbow trout, *Oncorhynchus mykiss*. In the first trial, apparent digestibility coefficient (ADC) was 92.4% for dry matter, 91.4% for crude protein, and 94.2% for crude lipid in rainbow trout. In the second trial, six diets were formulated to replace FO at 0% (CON), 20% (T20), 40% (T40), 60% (T60), 80% (T80), and 100% (T100) with SC in the rainbow trout (3.0 ± 0.4 g, mean ± SD) diet. After eight weeks’ feeding trial, weight gain (WG), specific growth rate (SGR), and feed efficiency (FE) of fish fed the T20 diet were significantly higher than those of fish fed other diets (*p* < 0.05). However, there were no significant differences in these parameters among those of fish fed CON, T40, T60, and T80 diets. Lysozyme activity of fish fed the T20 diet was significantly higher than those of fish fed other experimental diets (*p* < 0.05). After 10 days of disease challenge testing with pathogenic bacteria (*Lactococcus garvieae* 1 × 10^8^ CFU/mL), the cumulative survival rate of fish fed the T20 diet was significantly higher than those of fish fed the CON, T80, and T100 diets. Therefore, these results suggest dietary microalgae SC is well-digested and could replace up to 80% of fish oil in the diet of rainbow trout without negative effects on growth and immune responses.

## 1. Introduction

Commercial aquafeed formulation has historically relied on fish oil as the main source of dietary lipids. However, unfortunately, global fish oil production has witnessed a consistent downward trend since 2005, and therefore, there has been an ever-increasing demand for the consumption of fish oil [1]. Consequently, the price of fish oil has reached a historical peak and in the context of development of sustainable aquaculture as well as aquafeed, the search for promising alternative sources has received attention [2].

Researchers around the world have evaluated the potential of rapeseed oil [3], soybean oil [4], linseed oil [5], and poultry oil [6] as the lipid source in aquafeed formulation. Plant-based oil and animal by-products are comparatively cheaper than fish oils; however they have a relatively low/negligible amount of eicosatetraenoic acid (EPA) and docosahexaenoic acid (DHA) content, which limits their potential as fish oil alternatives [7]. Teleost fish should acquire EPA and DHA in the diet as a deficiency of these essential fatty acids could cause impaired immune and growth performance [1]. Thus, exploring those cost effective lipid source ingredients also rich in n-3 highly unsaturated fatty acids could be a more justified approach.

A variety of microalgae species have been an integral part of aquaculture industry as larvae food as well as for the early feeding stages of various fish and crustacean species [8]. Microalgae are known to be rich in essential amino acids and essential fatty acids, along with having relatively high protein and lipid content compared to several other plant-based ingredients [8]. However, interest in the use of microalgae in feed formulations has been recently highlighted.

Among various marine microalgae species, Thraustochytrid, Mucoreles and Ascomycetes are known to produce DHA, an unsaturated fatty acid [9,10]. *Thraustochytrids* have been reported to be suitable as a raw material for fish feed [11,12], as this group of microorganisms is capable of mass production and has a relatively high dry matter weight compared to other algae. *Schizochytrium* sp. (SC), a fast growing Thraustochytrid microalgae, has very high DHA content among microalgae and can optimize growth conditions by providing higher DHA (up to 60 percent enrichment). Additionally, this microalgae has lower omega-6 (especially docosapentaenoenoic acid, 22:6n-3), and higher omega-3/omega-6 ratios [13]. Therefore, SC has been shown to be effective in various forms as a potential substitute for fish meal in the feed formulation of channel catfish [14], gilthead bream [15], Atlantic salmon [10,13,16] and whiteleg shrimp [17]. In these studies, SC was shown to be a distinguished source of long chain polyunsaturated fatty acids (LCPUFA) such as DHA. However, little attention has been given to evaluate its potential as a fish oil substitute, particularly in species such as rainbow trout *Oncorhynchus mykiss* requiring high dietary lipid.

Rainbow trout is a freshwater carnivorous fish in the salmonid family, and is considered to be one of the world’s most popular aquaculture fish because of high tolerance in captivity and desirable taste [18]. The global aquaculture production of this species is continuously increasing and reached more than 900 thousand tons in 2020 [19]. The unsaturated fatty acid requirements of rainbow trout has already been established and a relatively high level of fish oil is incorporated in rainbow trout diet [20]. Therefore, the current study was carried out to evaluate the efficiency of microalgae SC as the dietary fish oil alternative in rainbow trout, *Oncorhynchus mykiss.*

## 2. Materials and Methods

### 2.1. Experimental Diets

In this experiment, the micro-algae, *Schizochytrium* sp. (SC) was provided by CJ Cheiljedang (Seoul, Republic of Korea) in an oily powder form. Dry matter, crude protein, crude lipid, EPA and DHA composition of SC are presented in Table 1. To assess the nutritional value of the SC, a digestibility trial was conducted followed by the growth trial.

In vivo digestibility was determined for SC product fed to rainbow trout. A reference diet containing practical ingredients and 0.1% chromium oxide was prepared. A batch of test diet contained 30% SC and 70% reference diet mash (combined on a dry matter basis) was prepared. Procedures for the digestibility trial are shown in Section 2.2.

For the growth trial, six isonitrogenous and isolipidic diets were formulated to compare substitution levels of FO in rainbow trout feed. The amounts 0%, 20%, 40%, 60%, 80%, and 100% of FO substitution with SC was performed to be the CON, T20, T40, T60, T80 and T100 diets, respectively. Formulations were similar to commercial feeds across the rainbow trout feed industry.

Dry ingredients were mixed by an electric mixer, and then fish oil and tap water were added to the suspension. After that, the mixed dough was pelleted using the screw-type pelleting machine (Baokyong Commercial Co., Busan, Korea) and dried at room temperature for approximately 48 h. After drying, the pellets were broken up to similar size, sieved to remove powder, sealed, and stored at −20 °C until use for feeding trial. Feed formulation and proximate composition for experimental diets are shown in Table 2, and analyzed fatty acids compositions are given in Table 3.

### 2.2. Digestibility Trial

The trial was conducted at Pukyong National University (PKNU), Busan, Republic of Korea. Domesticated rainbow trout (95.6 ± 2.1 g) were obtained from a private hatchery (Sangju-si, Gyeongbuk, Republic of Korea) for the digestibility trial. Rainbow trout were stocked into 250-L fiberglass tanks at the density of 30 fish per tank, each supplied with 10 L/min of filtered freshwater with a constant temperature (18 ± 1.0 °C). Experimental diets were fed to three replicate groups of fish in a completely randomized design to apparent satiation twice daily. 20–24 h post prandial, fish were anesthetized with tricaine methanosulfonate (MS-222, 100 mg/L). Feces were gently expelled from fish using light pressure on the abdomen near the vent, and collected in aluminum cups. Fish were stripped for feces twice per week for three weeks. Feces samples were stored frozen (−20 °C), dried at 75 °C for 24 h and analyzed for proximate composition and chromium oxide. Apparent digestibility coefficients (ADC) of diets, for dry matter, protein and lipid were calculated using the following formula described by Bureau et al. [21]:ADC_diet_ = 1 − ((F/D) × (Di/Fi))
where D = % nutrient of diet, F = % nutrient of feces,
Di = % digestion indicator of diet, Fi = % digestion indicator of feces
ADC_ingredient_ = ADCT + (((1 − s) DR)/s DI) × (ADCT − ADCR)
where ADCT = ADC of test diet, ADCR = ADC of reference diet, DR = % nutrient of reference diet,
DI = % nutrient of test ingredient, s = proportion of test ingredient in test diet (0.3)

### 2.3. Growth Trial

Rainbow trout were obtained from Sang-ju aquaculture farm (flow-through tanks) in the Rep. of Korea. Prior to the feeding trial, all the fish were fed the control diet for two weeks to acclimatized to the experimental conditions and facilities at PKNU. Three hundred sixty fish with initial average body weight of 3.0 ± 0.4 g (mean ± SD) were carefully stocked into each of the 18 tanks, resulting in 20 fish per tank (*n* = 3 tanks per treatment). Fish were fed twice a day (09:00 and 18:00 h) for 8 weeks at apparent satiation. The feeding trial was conducted using an indoor semi-recirculating system with eighteen 50-L tanks receiving filtered freshwater at the rate of 1.0 L/min. Supplementary aeration was provided to maintain dissolved oxygen levels near saturation. Water temperature (16.0 ± 1.0 °C), dissolved oxygen (7.12 ± 0.83 mg/L) and pH (7.5 ± 0.3) were maintained throughout the feeding trial. Fifty percent of water was exchanged everyday using filtered tap water; also, a photoperiod of 12 h light:12 h dark was employed throughout the experimental period. Fish were weighed and counted every two weeks throughout the duration of the study. Feeding was stopped for 24 h before each measurement to avoid inclusion of ingested feed in the weight measurement. At the end of the study, four fish per tank were removed, euthanized with tricaine methanesulfonate (MS-222, 250 mg/L) and frozen for proximate analysis. Five additional fish per tank were randomly selected, blood samples were collected from the caudal vein with heparinized syringes. Afterwards, plasma was separated by centrifugation at 10,000× *g* for 10 min and stored at −20 °C for determination of blood biochemical parameters and non-specific immune responses.

### 2.4. Biochemical and Immunological Analysis

Proximate composition analysis of the experimental diets, feces and fish were performed using the standard methods of AOAC [22]. Samples were dried at 105 °C to a constant weight to determine their moisture content. Ash content was determined by incineration at 550 °C (Method No. 930.05). Protein was determined using the Kjeldahl method (total nitrogen × 6.25) after acid digestion (Method No. 978.04), and crude lipid was ascertained by Soxhlet extraction (Method No. 930.09) using the Soxhlet system 1046 (Tecator AB, Hoganas, Sweden) after freeze-drying the samples for 20 h. The macro minerals including chromium were determined in ingredients, feeds and fecal samples using an Inductively Coupled Plasma Mass Spectrometer (Perkin-Elmer 3300, Waltham, MA, USA). The composition of fatty acid methyl esters was determined by gas chromatography (Trace GC, Theromo Finnigan, San Jose, CA, USA) with a flame ionization detector, equipped with a Carbowax 007 capillary column (30 m × 0.25 mm i.d., film thickness 0.25 μm, QUADREX, Bethany, CT, USA). Injector and detector temperatures were 250 °C. The column temperature was programmed from 100 °C to 220 °C at a rate of 5 °C min^−1^ and 220 °C to 240 °C at a rate of 3 °C min^−1^. Helium was used as the carrier gas. Fatty acids were identified by comparison with known standards. The plasma levels for glucose, total protein and activities of AST and ALT were measured using a chemical analyzer (Fuji DRI-CHEM 3500i, Fuji Photo Film Ltd. Tokyo, Japan). Superoxide dismutase (SOD) activity was measured by the superoxide radical dependent reaction inhibition rate of enzyme with WST-1 (Water Soluble Tetrazolium dye) substrate and xanthine oxidase using the SOD Assay Kit (Sigma-Aldrich 19160, Merck KGaA, Darmstadt, Germany) according to the manufacturer’s instructions. Each endpoint assay was monitored by absorbance at 450 nm (the absorbance wavelength for the colored product of WST-1 reaction with superoxide) after 20 min of reaction time at 37 °C. The percent inhibition was normalized by mg protein and expressed as SOD unit mg^−1^. After SOD analysis, SOD Kit waste was transferred to a chemical waste container for disposal in accordance with government regulations. Thereafter, safety management department of Pukyong National University was contacted for the disposal of the organic compounds at the disposal site. The lysozyme activity was analyzed as briefly described; test serum (0.1 mL) was added to 2 mL of a suspension of *Micrococcus lysodeikticus* (0.2 mg mL^−1^) in a 0.05 L sodium phosphate buffer (pH 6.2). The reactions were carried out at 20 °C and absorbance at 530 nm was measured between 0.5 min and 4.5 min using a spectrophotometer. The lysozyme activity unit was defined as the amount of enzyme producing a decrease in absorbance of 0.001/min.

### 2.5. Challenge Test

At the end of feeding trial, seven fish from each tank were randomly selected and redistributed based on their previous dietary treatment groups in 50-L capacity aquaria. The pathogenic bacterium, *Lactococcus garvieae*, was obtained from the Department of Biotechnology, Pukyong National University, Busan, Republic of Korea. Each fish was injected intraperitoneally with 0.1 mL of *Lactococcus garvieae* at 1 × 10^8^ CFU/mL. Fish mortality was recorded daily up to 10 days. Water temperature was maintained at 18 ± 0.5 °C (mean ± SD) during the bacterial challenge test.

### 2.6. Statistical Analysis

All data were checked for normality and homogeneity of variance using Kolmogorov–Smirnov and Levene’s test, respectively. Data were analyzed by one-way ANOVA (Statistix 3.1, Analytical Software, St. Paul, MN, USA) to test the effects of the dietary treatments. When a significant treatment effect was observed, a Least Significant Difference (LSD) test was used to compare means. Treatment effects were considered at *p* < 0.05 level of significance.

## 3. Results

### 3.1. Growth Performance

Growth performance of juvenile rainbow trout fed different experiential diets for eight weeks is summarized in Table 4. Fish in the experiment responded differently to various fish oil replacement diets. Weight gain, SGR, FE, and PER of rainbow trout fed the T20 diet was significantly higher than those of fish fed CON, T40, T60, T80, and T100 diets. However, beyond the 20% fish oil replacement with microalgae SC, growth performance tended to gradually decline. Data for WG, SGR, FE, and PER showed no significant differences in these parameters among those fish fed CON, T40, T60, T80, and also T60, T80, or T100 diets. The overall survival rate was recorded beyond 93%; however, no significant difference could be recorded among different dietary treatments.

### 3.2. Whole-Body Proximate and Fatty Acid Composition

Table 5 shows the whole body proximate and fatty acids composition of juvenile trout fed different experimental diets for eight weeks. Although numerical differences were recorded in crude protein, lipid, ash, and crude moisture content, but no clear trends could be observed in terms of the effects of fish oil replacement on the fish whole body nutrient profile. On the other hand, significant effects of graded level of microalgae inclusion and fish oil replacement on the fatty acids profile could be clearly observed. Whole body EPA content tend to significantly decrease but DHA tend to significantly increase with the subsequent increase in the graded level of dietary microalgae inclusion as the fish oil alternative.

### 3.3. Hematologic Analysis

Table 6 shows the hematologic characteristics of juvenile rainbow trout fed different experimental diets for eight weeks. Although numerical differences were recorded in the GOT, GPT, GLU, and TP content of rainbow trout fed different experimental diets, no clear pattern could be observed in terms of effects of fish oil replacement on these aforementioned parameters.

### 3.4. Non-Specific Immune Responses

Superoxide dismutase and lysozyme activity of fish fed different experimental diets have been summarized in Figure 1. Lysozyme activity of rainbow trout fed the T20 diet was significantly higher than those of fish fed other experimental diets. However, there was no significant difference in the lysozyme activity of fish fed CON, T40, T60 or those fed T80 and T100 diets. Although numerical differences were recorded in SOD content of experimental fish fed different experimental diets, no clear trend could be drawn.

### 3.5. Disease Resistance against Bacterial Infection

Figure 2 shows the cumulative survival rate of rainbow trout after 10 days of challenge test with pathogenic bacteria. At the end of the fourth day, a significantly lower survival rate was recorded among the fish fed fish oil-based (control) diet compared to other (T20~T100) diets. However, there was no significant difference in the cumulative survival rate among rainbow trout infected with the bacteria *Lactococcus garvieae* fed T20, T40, T60, T80, or T100 diets.

## 4. Discussion

The last few years have witnessed a great deal of interest in the commercial exploitation of microalgae in the biofuel, pharamaceutical, food, and animal feed industries [23]. A handful of studies carried out recently [23,24,25,26], suggested the potential of microalgae species as the protein and/or lipid sources in the commercial aquafeed formulations. Furthermore, the current experiment clearly demonstrates the efficacy of microalgae SC to replace up to 80% of fish oil in the diet of rainbow trout. Experimental diets including fish oil-based (control) and those that were replaced with SC (T20~T100), were well accepted and consumed by the rainbow trout. Analyzed nutrient content of different experimental diets closely resembled our original targeted formulation (Table 2 and Table 3). Observations from the nutrient digestibility study were encouraging (Table 1) and concurrent with other parameters analyzed in this experiment. Previous studies were conducted on microalgae and unicellular species in different aquaculture species, but among them only a few published digestibility data. In a recent study, three species of microalgae (*Chlorella vulgaris, Scenedesmus dimorphus, and Nannochloropsis gadi-tana*) and a cyanobacterium (*Arthrospira maxima*) were tested in the diet of Nile tilapia (*Oreochromis niloticus*) and African catfish (*Clarus gariepinus*). In both fish species, *A. maxima* was reported to exhibit the highest apparent digestibility coefficient (81.4~82.5%), followed by other three microalgae species [27]. These researchers also reported fat ADCs ranged between 65.1 and 89.1% for these aforementioned microalgae and cyanobacteria species, whereas in the present study, ADC was 92.4% for dry matter, 91.4% for crude protein and 94.2% for crude lipid. It is worth noting an existing hypothesis in this regard: the cell walls of unicellular sources might hinder nutrient accessibility, leading to a decreased nutrient digestibility, whereas there have been reports of great diversity in the cell wall structures of microalgae and cyanobacteria, from peptidoglycan cell walls to cellulosic cell walls [28,29]. It is commonly assumed, but without clear experimental evidence, that peptidoglycan cell walls are softer than cellulose-based cell walls. Additionally, most fish lack the enzymes needed for cellulose degradation [30]. Nevertheless, a cross comparison of these published digestibility reports with our observations from the current experiment, microalgae SC appeared to be highly digestible (Table 1) in the diet of carnivorous fish such as rainbow trout.

Growth performance from the current study compares favorably with similar studies [31,32] carried out previously with the same species. Our observations are in agreement with those of earlier reports, suggesting a fish oil replacement level of 80~90% using alternative plant/animal lipid sources without compromising the growth performance in Rainbow trout [24]. Interestingly, fish in the experiment exhibited an improved growth performance (WG, SGR, and FE) at 20% fish oil replacement (T20), afterwards this gradually declined (Table 4). On the other hand, dietary algae inclusion as the fish oil substitute did not have any significant effect on the liver weight (HSI) gain. Likewise, European seabass fed graded levels of *T. lutea* did not exhibit any changes in major biometry and HSI [33,34], and also reported unchanged value of HSI in the same fish species due to microalgae inclusion. The possible reason for the gradual decline in the fish growth beyond 20% of fish oil replacement could be due to the hard/indigestible cell wall of microalgae. In a recent study, [27] investigated the cell wall hardness and their effects on digestive characteristics of three different microalgae species. These researchers hypothesized a better efficiency of herbivorous fish to access and therefore digest unicellular proteins compared to omnivorous fish species due to the differences in the robustness of the cell walls of microalgae. However, [25] could obtain a complete substitution of fish oil using the microalgae *Schizochytrium* sp. in the diet of tilapia. Rainbow trout is well-known as a carnivorous fish species which has limited efficiency to digest microalgae. As a result, up to 40 g/kg (T80) of microalgae *Schizochytrium* sp. inclusion at the expense of dietary fish oil could be the optimum. Further studies in this regard are recommended to clearly understand the mechanism hindering the inclusion of microalgae at higher level in commercial fish feed formulation, particularly for carnivorous species such as rainbow trout.

Dietary sources as well as lipid profile have frequently been reported to affect the product quality [35,36,37,38] and the growth performance and health of farmed fishes [24]. In addition, product quality of farmed fishes is measured by their nutrient profile, physical appearance, and the acceptance of the final consumer [39]. The significance of polyunsaturated fatty acids profile in fish muscle is of prime importance with regard to human health benefits associated with the farmed fish consumption [40]. Our observations from the whole body composition (Table 5), showed insignificant effects of dietary fish oil substitution with microalgae in rainbow trout. It appears that microalgae in fish diets may have negligible effects on the fish whole body crude protein, lipid, ash, and crude moisture content. Similarly, for instance, [23] could not find any significant effects of dietary microalgae on the whole body proximate composition of salmon and common carp. Likewise, no major changes in muscle lipid due to graded dietary levels of *Tisochrysis lutea* were observed by [33] in European seabass. On the other hand, there have been reports of adverse effects of fish oil replacement on the EPA and DHA content of fish body and altered liver fatty acids metabolism in fish species such as gilthead seabream [41], pikeperch [42] and Atlantic salmon [3,43]. In the current experiment, dietary fatty acids profile well mirrored in the fish whole body fatty acid composition. Saturated fatty acids and multi saturated fatty acids were unaffected by the fish oil substitution with the graded levels of microalgae inclusion. On the other hand, EPA tend to decrease but DHA tend to increase with a subsequent increase in the dietary microalgae inclusion. Nevertheless, data for total EPA and DHA content (Table 5) showed no adverse effects of dietary fish oil substitution with the microalgae SC inclusion in rainbow trout. Overall, it appeared that fish oil could be replaced with dietary microalgae without any adverse effects on fish whole body composition and without compromising the fatty acid profile in the rainbow trout.

Hematologic characteristics are believed to indicate the health status of animal including aquaculture species [44]. Any stress caused by physiochemical parameters or poor nutrition were reported to have a negative impact on heamotological characteristics such as glutamic oxaloacetic transaminase (GOT), glutamic pyruvic transaminase (GPT), total protein, and blood glucose level [45]. Observations from the current experiment (Table 6) showed no significant effects of fish oil replacement with the graded levels of dietary microalgae on the hematologic characteristics of the rainbow trout, suggesting no adverse effects on fish health.

In the context of growing stress due to intensification and environmental pollution, feed is believed to play a central role in maintaining health and ensuring optimum growth and survival of aquaculture species [46]. Non-specific immunity of fish is the fundamental defense system, playing key role in acquiring immune responses and homeostasis [46]. Super oxide dismutase (SOD) is an antioxidant enzyme well-known to control the reactive oxygen species in cells and catalyze the dismutation of hydrogen peroxide and oxygen from superoxide radicle. In addition, lysozyme is a mucolytic enzyme of leucocitic origin and has frequently been used as the potential indicator of nonsepecific immunity, known for medicating protection against microbial invasion [47]. Increased lysozyme activity is an indication of improved immune response in fish [48,49,50]. In the current experiment, no clear trend could be drawn in SOD activity (Figure 1). However, observations with the lysozyme activity (Figure 1) clearly showed the beneficial effects of microalgal inclusion as well as no adverse effects of fish oil replacement on the non-specific immunity in rainbow trout.

As far as we know, this is the first report demonstrating the effects of fish oil replacement and graded levels of microalgae inclusion on disease resistance in rainbow trout. *Lactococcus garvieae* is a well-known disease bacteria in fish causing important economic losses in rainbow trout aquaculture [51]. In this experiment, results from the disease challenge test well corroborated with other observed parameters. A higher survival rate and disease resistance against the pathogenic bacteria species, *Lactococcus garvieae* (Figure 2) could be observed among rainbow trout fed graded level of microalgae as the fish oil substitute. This could be explained by the higher deposition of LCPUFA in fish (Table 5) as a result of SC supplementation. The same results were reported by [25] when Nile tilapia was fed by SC. LCPUFA were reported to improve immune responses in fish [52] and this could be a reason for our observations. However, further studies regarding the effects of SC on fish immune responses are required. As the concept of preventive health management is gaining momentum and the use of antibiotics for disease treatment/prevention have become unpopular [44], the inclusion of dietary microalgae could open a potential avenue in the field of functional feed formulation for commercial aquaculture species.

## 5. Conclusions

In conclusion, microalgae *Schizochytrium* sp. is a digestible ingredient in the rainbow trout diet. In the feed formulation of this carnivorous fish species, up to 80% of fish oil could be replaced using microalgae without any adverse effects on the fish growth performance, whole body nutrient profile and hematologic characteristics. However, the highest growth performance was observed at 20% replacement of fish oil with microalgae. The inclusion of graded level of microalgae *Schizochytrium* sp. could also improve the non-specific immunity and disease resistance against bacterial infection in rainbow trout.

## Figures and Tables

**Figure 1 animals-12-01220-f001:**
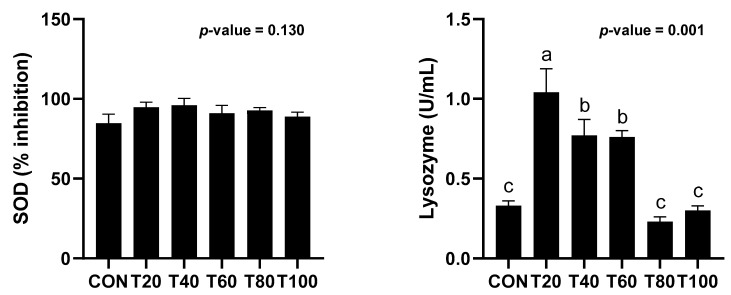
Non-specific immune responses of juvenile rainbow trout fed experimental diets for 8 weeks. Values are means from triplicate groups of fish where the values in each row with different superscripts are significantly different (*p* < 0.05).

**Figure 2 animals-12-01220-f002:**
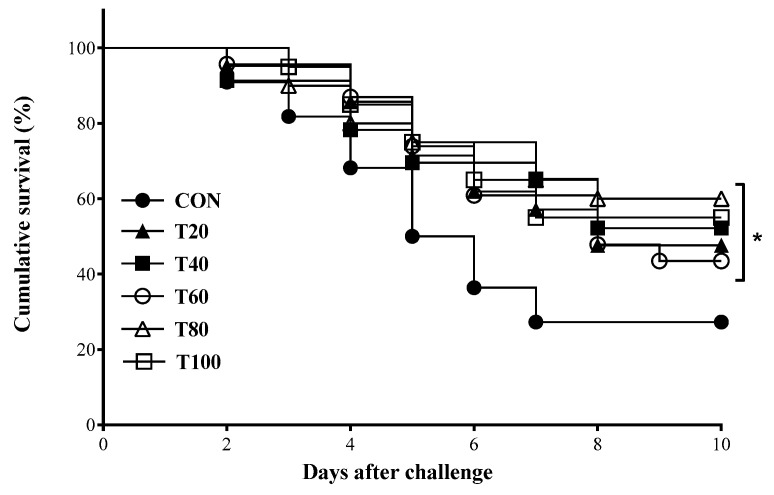
Cumulative survival rate after challenge with *Lactococcus garvieae* for 10 days in rainbow trout fed the experimental diets for 8 weeks. *p*-value: 0.038. Values are means from triplicate groups of fish where asterisks (*) indicates significant differences among groups (*p* < 0.05).

**Table 1 animals-12-01220-t001:** Nutrients composition and apparent digestibility coefficients of *Schizochytrium* sp. (% dry matter basis).

	Nutrient Composition	Apparent Digestibility
Dry matter	99.7	92.4
Crude protein	20.9	91.4
Crude lipid	55.3	94.2
Fatty acids (% of total fatty acids)		
∑ SFA ^1^	42.9	94.3
∑ MUFA ^2^	1.18	95.8
C18:2n-6	19.2	93.8
C18:3n-6	0.33	100
C18:3n-3	0.43	98.5
C20:2n-6	0.00	-
C20:3n-6	0.52	99.0
C20:4n-6	2.35	94.5
C20:5n-3 (EPA)	0.20	100
C22:6n-3 (DHA)	29.1	92.8

^1^ saturated fatty acids include 10:0, 12:0, 13:0, 14:0, 15:0, 16:0, 17:0, 18:0, and 20:0. ^2^ monounsaturated fatty acids include 16:1, 18:1, and 20:1.

**Table 2 animals-12-01220-t002:** Ingredient composition of the experimental diets fed to rainbow trout juveniles over an eight-week growth trial (% of DM basis).

Ingredients	Diets (%)					
	CON	T20	T40	T60	T80	T100
Fish oil ^1^	6.80	5.43	4.07	2.71	1.36	0.00
Micro algae ^2^		1.00	-	-	-	-
		-	2.01	-	-	-
		-	-	3.01	-	-
		-	-	-	4.02	-
		-	-	-	-	5.01
Fish meal (Chile) ^3^	20.00	20.00	20.00	20.00	20.00	20.00
Soybean meal ^3^	19.00	18.55	18.20	17.80	17.20	16.90
Poultry by-product ^3^	16.12	16.12	16.12	16.12	16.12	16.12
Blood meal ^3^	5.70	5.70	5.70	5.70	5.70	5.70
Meat and bone meal ^3^	9.00	9.00	9.00	9.00	9.00	9.00
Wheat gluten meal ^3^	5.00	5.00	5.00	5.00	5.00	5.00
Wheat flour ^3^	12.90	12.87	12.73	12.56	12.76	12.58
Soybean oil ^3^	0.00	0.82	1.63	2.52	3.24	4.06
Methionine ^3^	0.43	0.44	0.44	0.44	0.45	0.45
Lysine ^3^	0.51	0.53	0.56	0.59	0.61	0.64
Choline ^3^	0.54	0.54	0.54	0.54	0.54	0.54
Vitamin premix ^4^	2.00	2.00	2.00	2.00	2.00	2.00
Mineral premix ^5^	2.00	2.00	2.00	2.00	2.00	2.00
Total	100.0	100.0	100.0	100.0	100.0	100.0

^1^ Provided by E-Wha Oil Co., Ltd., Busan, Korea; ^2^ *Schizochytrium* sp. powder provided by Cheiljedang Oil Co., Ltd., Seoul, Korea; ^3^ Provided by Su-hyup Feed Co., Ltd., Uiryeong, Korea; ^4^ Contains (as mg kg^−1^ in diets): Ascorbic acid, 300; DL-Calcium pantothenate, 150; Choline bitatrate, 3000; Inositol, 150; Menadione, 6; Niacin, 150; Pyridoxine HCl, 15; Riboflavin, 30; Thiamine mononitrate,15; dl-a-Tocopherol acetate, 201; Retinyl acetate, 6; Biotin, 1.5; Folic acid, 5.4; B12, 0.06; ^5^ Contains (as mg kg^−1^ in diets): NaCl, 437.4; MgSO_4_·7H_2_O, 1379.8; ZnSO_4_·7H_2_O, 226.4; Fe-Citrate, 299; MnSO_4_, 0.016; FeSO_4_, 0.0378; CuSO_4_, 0.00033; Calciumiodate, 0.0006; MgO, 0.00135; NaSeO_3_, 0.00025.

**Table 3 animals-12-01220-t003:** Proximate composition (% of DM basis) and fatty acids composition (% of total fatty acid) of experimental diets.

	Diets (%)					
	CON	T20	T40	T60	T80	T100
Moisture	9.36	8.50	8.81	9.05	8.71	8.54
Ash	11.18	11.17	11.16	11.41	11.14	11.02
Crude Protein	50.46	49.57	49.69	48.71	49.81	49.84
Crude Lipid	12.00	11.92	11.54	12.6	12.72	12.16
Fatty acid (% of total fatty acids)						
∑ SFA ^1^	56.9	57.9	57.6	59.0	57.7	56.5
∑ MUFA ^2^	20.9	19.9	18.7	18.2	17.4	16.5
C18:2n-6	4.24	4.27	4.60	4.52	6.90	7.73
C18:3n-6	0.37	0.40	0.39	0.41	0.41	0.42
C18:3n-3	0.28	0.27	0.25	0.26	0.31	0.33
C20:2n-6	0.16	0.15	0.13	0.13	0.10	0.11
C20:3n-6	0.27	0.30	0.24	0.25	0.25	0.25
C20:4n-6	1.42	1.39	1.33	1.32	1.25	1.19
C20:5n-3 (EPA)	9.49	7.36	6.53	4.48	2.96	1.43
C22:6n-3 (DHA)	6.45	8.65	10.5	11.9	13.4	16.1
EPA + DHA	15.9	16.0	17.0	16.4	16.3	17.5

^1^ saturated fatty acids include 10:0, 12:0, 13:0, 14:0, 15:0, 16:0, 17:0, 18:0, and 20:0. ^2^ monounsaturated fatty acids include 16:1, 18:1, and 20:1.

**Table 4 animals-12-01220-t004:** Growth performance of juvenile rainbow trout fed the experimental diets for 8 weeks ^1^.

	CON	T20	T40	T60	T80	T100	Pooled SEM	*p* Value
IBW ^2^	3.15	3.17	3.15	3.15	3.15	3.17	0.02	0.998
FBW ^3^	9.10 ^b^	10.9 ^a^	9.01 ^b^	8.70 ^bc^	8.42 ^bc^	7.28 ^c^	0.50	<0.001
WG (%) ^4^	191 ^b^	246 ^a^	186 ^b^	176 ^bc^	167 ^bc^	130 ^c^	13.6	<0.001
SGR (%/day) ^5^	1.94 ^b^	2.25 ^a^	1.91 ^b^	1.85 ^bc^	1.79 ^bc^	1.51 ^c^	0.09	<0.001
FE (%) ^6^	110 ^b^	139 ^a^	105 ^b^	98.8 ^bc^	94.7 ^bc^	73.6 ^c^	7.70	<0.001
FCR ^7^	0.91 ^b^	0.72 ^c^	0.95 ^b^	1.01 ^ab^	1.06 ^ab^	1.36 ^a^	0.04	<0.001
PER ^8^	2.16 ^b^	2.73 ^a^	2.11 ^b^	1.98 ^bc^	1.93 ^bc^	1.50 ^c^	0.14	<0.001
Survival (%) ^9^	93.3	96.7	93.3	98.3	96.7	98.3	0.85	0.174
HSI (%) ^10^	1.49 ^ns^	1.40	1.55	1.47	1.50	1.12	0.05	0.162

^1^ Values are means from triplicate groups of fish where the values in each row with different superscripts are significantly different (*p* < 0.05); ^2^ Initial weight (IW g/fish); ^3^ Final weight (FW g/fish); ^4^ Weight gain (WG, %) = (final weight − initial weight) × 100/initial weight; ^5^ Specific growth rate (SGR, %/day) = (ln final weight − ln initial weight) × 100/days; ^6^ Feed Efficiency (FE, %) = (wet weight gain/dry feed intake) × 100; ^7^ Feed conversion ratio (FCR) = (dry feed intake/wet weight gain), ^8^ Protein efficiency ratio (PER) = (wet weight gain/protein intake), ^9^ Survival rate (%) = (total fish − dead fish) × 100/total fish; ^10^ Hematosomatic index (HSI, %) = liver wt. × 100/body weight.

**Table 5 animals-12-01220-t005:** Whole body proximate and fatty acids composition (%) of juvenile rainbow trout fed experimental diets for 8 weeks ^1^.

	CON	T20	T40	T60	T80	T100	Pooled SEM	*p* Value
Crude Protein	74.5	75.1	74.8	74.7	74.4	77.0	0.41	0.783
Crude Lipid	12.6	11.6	13.7	13.4	14.1	11.9	0.63	0.114
Crude Ash	12.6	12.8	13.0	12.5	13.4	13.4	0.11	0.366
Moisture	76.4	78.4	76.3	78.5	78.7	78.7	0.49	0.076
Fatty acids (% of total fatty acids)								
∑ SFA ^2^	30.0	30.7	30.5	30.6	30.3	30.5	0.17	0.893
∑ MUFA ^3^	22.5	22.5	22.7	22.0	22.1	22.0	0.23	0.730
C18:2n-6	6.47	6.50	6.49	6.60	6.66	6.58	0.04	0.326
C18:3n-6	0.45	0.44	0.42	0.45	0.45	0.43	0.01	0.445
C18:3n-3	1.58	1.60	1.61	1.66	1.62	1.67	0.02	0.638
C20:2n-6	0.40	0.39	0.35	0.37	0.37	0.34	0.01	0.257
C20:3n-6	0.94	0.97	0.92	0.93	0.93	0.93	0.03	0.330
C20:4n-6	0.98	0.92	0.94	0.94	0.92	0.90	0.04	0.159
C20:5n-3 (EPA)	6.23 ^a^	6.15 ^a^	5.33 ^b^	4.74 ^c^	3.90 ^d^	2.38 ^e^	0.52	<0.001
C22:6n-3 (DHA)	14.3 ^c^	15.8 ^bc^	15.9 ^bc^	16.7 ^b^	18.0 ^ab^	19.2 ^a^	0.49	<0.001
EPA + DHA	20.6	22.0	21.3	21.5	22.0	21.6	0.16	0.119

^1^ Values are means from triplicate groups of fish where the values in each row with different superscripts are significantly different (*p* < 0.05); ^2^ saturated fatty acids include 10:0, 12:0, 13:0, 14:0, 15:0, 16:0, 17:0, 18:0, and 20:0. ^3^ monounsaturated fatty acids include 16:1, 18:1, and 20:1.

**Table 6 animals-12-01220-t006:** Hematologic analysis of juvenile rainbow trout fed experimental diets for 8 weeks ^1^.

	CON	T20	T40	T60	T80	T100	Pooled SEM	*p* Value
GOT (U/L ^2^	30.9	38.3	32.7	34.8	38.8	39.4	1.39	0.674
GPT (U/L) ^3^	9.00	11.7	10.3	10.7	11.0	11.7	0.45	0.679
GLU (mg/dL) ^4^	81.7	80.3	82.0	87.0	88.3	87.3	1.11	0.762
TP (g/dL) ^5^	3.37	3.50	3.37	3.33	3.67	3.57	0.04	0.671

^1^ Values are means from triplicate groups of fish where the values in each row with different superscripts are significantly different (*p* < 0.05); ^2^ Glutamic oxaloacetic transaminase (U/L); ^3^ Glutamic oxaloacetic transaminase (U/L); ^4^ Glucose (mg/dL); ^5^ Total protein (g/dL).

## Data Availability

The data that support the findings of this study are available on request from the corresponding author. The data are not publicly available due to privacy or ethical restrictions.

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
