# Peer review of "Partial Substitution of Fish Oil with Microalgae (Schizochytrium sp.) Can Improve Growth Performance, Nonspecific Immunity and Disease Resistance in Rainbow Trout, Oncorhynchus mykiss"

_animals, 2022, doi:10.3390/ani12091220_

Round 1

Reviewer 1 Report

General opinion: Presented study has the potential to be an important element of science. However, I did observe few extremely significant errors both in the description of the methodology and in the experiment design itself:

  1. Materials and methods L 138: If you refer to AOAC in case of proximate composition analysis please mention specific methods of AOAC by their number
  2. Materials and methods L 176: In statistical analysis there is no information about the testing towards the normality distribution of data. How do you know that one-way ANOVA is the right test to use in statistical analysis?
  3. Table 2 L 184: As the levels of fish oil in experimental diets decrease, the level of soybean oil increases significantly. How authors can distinguish that the effect results from the addition of microalgae and not soybean oil, which – importantly – is not included in the control diet. In my opinion, this seems to be serious methodological error. To avoid such errors, authors should add soybean oil in the control diet to be sure, that it will be not another experimental factor.
  4. When using innovative feed materials, it is extremely important to perform complete analyses in terms of nutrient composition. It is unacceptable to focus only on dry matter, crude protein and crude fat. Protein and fat together account for about 76% of the chemical composition of this feed material. What is the remaining 24%?

Additionaly, in my opinion manuscript should be edited by native speaker in terms of the correctness of the language, avoiding the repetition of words, style, and above all, for the text to be more understandable for the reader.

Author Response

Dear Reviewers and Editor,

We sincerely thank you for your efforts in improving the quality of this manuscript through your thoughtful and considerate comments and suggestions. We carefully considered your comments one-by-one, performed the necessary changes in the manuscript (track-changes on) and responded to each comment here (bold font). If there are further concerns or suggestions, please don’t hesitate to let us know.

Sincerely yours,

Authors

Comments and Suggestions for Authors

General opinion: Presented study has the potential to be an important element of science. However, I did observe few extremely significant errors both in the description of the methodology and in the experiment design itself:

1. Materials and methods L 138: If you refer to AOAC in case of proximate composition analysis please mention specific methods of AOAC by their number

Response: According to the reviewer’s suggestion, method number was added to the manuscript.

2. Materials and methods L 176: In statistical analysis there is no information about the testing towards the normality distribution of data. How do you know that one-way ANOVA is the right test to use in statistical analysis?

Response: We appreciate your thorough review and giving us an opportunity to clarify regarding statistical analysis. All data were checked for normality and homogeneity of variance using Kolmogo-rov–Smirnov and Levene's test, respectively. This was added to the manuscript. In this experiment, the effects of only one variable “lipid source” was tested whereas all other parameters in the feed formulation as well as in the conducting trial were maintained similar for all the treatments and one way ANOVA was employed.

3. Table 2 L 184: As the levels of fish oil in experimental diets decrease, the level of soybean oil increases significantly. How authors can distinguish that the effect results from the addition of microalgae and not soybean oil, which – importantly – is not included in the control diet. In my opinion, this seems to be serious methodological error. To avoid such errors, authors should add soybean oil in the control diet to be sure, that it will be not another experimental factor.

Response: Thanks again for such an observation and highlighting a pertinent point. Balancing the fatty acids profile including EPA and DHA is often a major challenge in Fish oil replacement studies. Likewise, in order to balance the amino acids profile a minor balance in other protein ingredients are employed in fish meal replacement studies. In the current experiment, Fraction of Soybean oil was added and gradually increased in order to ensure the balance of fatty acids profile.

4. When using innovative feed materials, it is extremely important to perform complete analyses in terms of nutrient composition. It is unacceptable to focus only on dry matter, crude protein and crude fat. Protein and fat together account for about 76% of the chemical composition of this feed material. What is the remaining 24%?

Response: Proximate composition analysis include protein, fat, moisture and ash content, whereas carbohydrate content could be calculated accordingly, as you could see the detail in table 3 of our feed formulation. Here the new feed material viz. Microalgae is primarily known for its fat, protein and fatty acids content in the context of viable alternative of fish oil. Accordingly, these aforementioned nutrients were analysed and summarised in the table. Remaining 24% accounts for moisture and ash content. Hope this response was satisfactory to the reviewer’s concern.

Additionaly, in my opinion manuscript should be edited by native speaker in terms of the correctness of the language, avoiding the repetition of words, style, and above all, for the text to be more understandable for the reader.

Response: We highly appreciate your efforts and comments to further improve the authenticity and readability of our publications. Accordingly, we have made necessary changes throughout the paper for your further review.

Reviewer 2 Report

The manuscript of Seunghan Lee and colleagues focuses on a very current topic a lot treated in the last years, due to the need for fish oil and fish meal replacement in the aquaculture field. Despite this, the experimental trials conducted by this research group focused on something new that can give important relapses to this research field. The study was conducted in a good manner and some key results were obtained, which I suggest the Authors refer to in their manuscript, better considering their data in the result section and the discussion and conclusion too, as suggested in this report. 

Abstract

This section is too long, both from a reader's point of view and from Animals Journal rules (limit is 200 words).

Keywords

Please try to substitute words already present in the Title with related ones.

Introduction

Please insert something about Lactococcus garvieae that you used in this study as a challenge. How is related to Oncorhynchus mykiss and in general with farmed species.

Material and Methods

Line 75: ..was provided in what form?

I suggest placing Tables 1, 2, and 3 after section 2.1

Line 98: data on length hasn't been collected?

Line 99-101: data on dissolved oxygen and photoperiod hasn't been collected?

Can the Authors explain why did they choose using different temperatures for the two experiments (digestibility and growth)? How this high difference is linked to the size of specimens or the experimental design of each trial?

Results

Table 4 highlights in my opinion a constant decrease trend of almost all the parameters related to the efficiency of the experimental feed. Only T20 seems to have a positive effect compared to control, but the other ones were not convenient from a growth point of view. Please reconsider your comments taking into account that. (I see your discussion about that, that are shareable hypothesis, so you have to consider the decreasing trend also in the result)

Subsection 3.2 contains some discussion contents, please leave the comments for the discussion section only. The result section should focus only on the report of your data.

Figure 2, in the same manner as Table 4 comments, shows interesting survival data from experimental diets of fish, that seem to have more resistance to pathogen effect, compared to control. From the graph, I mean over two times more. Why the Authors didn't give resonance to this data?

Discussion

Line 273: Why 80% if you used T100 with total replacement of fish oil? I don't understand if I'm wrong.

Conclusion

Line 382: linking to Table 4 data and your discussion about that, I don't know if highly digestible is correct. If you remind of some difference in growth rate with the increase of microalgal inclusion, due to the difficulty of a predatory fish to manage the vegetable cell wall, this sentence goes a little in contrast.

Lines 383-386: I suggest to the Authors give more resonance to their key results in the conclusion separately, for example highlighting the best growth performance on 20% of FO replacement, etc. This sentence is in my opinion too generic giving a 20-80% range and doesn't reflect the manuscript contents.

Best regards

The Reviewer

Author Response

Dear Reviewers and Editor,

We sincerely thank you for your efforts in improving the quality of this manuscript through your thoughtful and considerate comments and suggestions. We carefully considered your comments one-by-one, performed the necessary changes in the manuscript (track-changes on) and responded to each comment here (bold font). If there are further concerns or suggestions, please don’t hesitate to let us know.

Sincerely yours,

Authors

Reviewer2

The manuscript of Seunghan Lee and colleagues focuses on a very current topic a lot treated in the last years, due to the need for fish oil and fish meal replacement in the aquaculture field. Despite this, the experimental trials conducted by this research group focused on something new that can give important relapses to this research field. The study was conducted in a good manner and some key results were obtained, which I suggest the Authors refer to in their manuscript, better considering their data in the result section and the discussion and conclusion too, as suggested in this report. 

Abstract

This section is too long, both from a reader's point of view and from Animals Journal rules (limit is 200 words).

Response: Accordingly, we have modified the abstract part, limited to 200 words as per the journal format.

Keywords

Please try to substitute words already present in the Title with related ones.

Response: The keywords were modified accordingly.

Introduction

Please insert something about Lactococcus garvieae that you used in this study as a challenge. How is related to Oncorhynchus mykiss and in general with farmed species.

Response: A statement has been added to the challenge test paragraph in the discussion part line 390-392.

Material and Methods

Line 75: ..was provided in what form?

Response: In oily powder form. This was clarified in the text file.

I suggest placing Tables 1, 2, and 3 after section 2.1

Response: According to the reviewer’s suggestion, we have placed these tables after section 2.1.

Line 98: data on length hasn't been collected?

Response: Due to the objectives of this project, our main focus was the effects of fish oil replacement on biomass production, immune responses, etc. That’s why we did not measure fish length.

Line 99-101: data on dissolved oxygen and photoperiod hasn't been collected?

Response: Thanks for pointing out these important factors. We have data for dissolved oxygen and photoperiod. Actually, in line 148 we already mentioned that:” a photoperiod of 12h light: 12h dark was employed”. Also, we added the dissolved oxygen data to the manuscript (line 146).

Can the Authors explain why did they choose using different temperatures for the two experiments (digestibility and growth)? How this high difference is linked to the size of specimens or the experimental design of each trial?

Response: Thanks for the attention to details. We made a typo in presenting the average temperature of the growth trial. The temperature for our growth trail was 16±1. This was corrected in the manuscript.

Results

Table 4 highlights in my opinion a constant decrease trend of almost all the parameters related to the efficiency of the experimental feed. Only T20 seems to have a positive effect compared to control, but the other ones were not convenient from a growth point of view. Please reconsider your comments taking into account that. (I see your discussion about that, that are shareable hypothesis, so you have to consider the decreasing trend also in the result)

Response: We agree with the reviewer. In the results section line 211-212 the decline of growth performance parameters was mentioned.

Subsection 3.2 contains some discussion contents, please leave the comments for the discussion section only. The result section should focus only on the report of your data.

Response: According to the reviewer’s comment, we have modified this part.

Figure 2, in the same manner as Table 4 comments, shows interesting survival data from experimental diets of fish, that seem to have more resistance to pathogen effect, compared to control. From the graph, I mean over two times more. Why the Authors didn't give resonance to this data?

Response: In lines 262-264 we have clearly mentioned that the control group showed the lowest survival among all other diets.

Discussion

Line 273: Why 80% if you used T100 with total replacement of fish oil? I don't understand if I'm wrong.

Response: The reason in because up to 80% replacement of fish oil did not cause negative effects on growth, immune responses and body composition. At 100%, growth was significantly reduced, so we don’t suggest replacement at 100%.

Conclusion

Line 382: linking to Table 4 data and your discussion about that, I don't know if highly digestible is correct. If you remind of some difference in growth rate with the increase of microalgal inclusion, due to the difficulty of a predatory fish to manage the vegetable cell wall, this sentence goes a little in contrast.

Response: We understand the reviewer’s concern. Actually, we used the term “highly digestible” based on our results for digestibility test in Table 1 (91% protein and 94% lipid digestibility). However, we agree with the reviewer that digestibility could be a reason for growth depression in Table 4. So we deleted “highly” to make this sentence less strong.

Lines 383-386: I suggest to the Authors give more resonance to their key results in the conclusion separately, for example highlighting the best growth performance on 20% of FO replacement, etc. This sentence is in my opinion too generic giving a 20-80% range and doesn't reflect the manuscript contents.

Response: According to the reviewer’s suggestion, a sentence was added and this part was modified.

Reviewer 3 Report

The paper "Partial substitution of fish oil with microalgae (Schizochytrium sp.) can improve growth performance, nonspecific immunity and disease resistance in rainbow trout, Oncorhynchus mykiss" was designed to evaluate whether dietary microalgae could be used as a substitute for fish oil in rainbow trout rearing. It was found that dietary microalgae had good digestibility and could replace up to 80% of fish oil without affecting the growth performance, whole-body nutritional integrity and health of the carnivorous fish rainbow trout.

  1. The abstract is roughly divided into background, purpose and significance, methods, results and conclusions. The results of this abstract are too descriptive, the description of the research background is lacking, and the statement of the conclusions is not clear enough.
  2. Why are digestion experiments performed separately from other experiments?
  3. Please unify the capitalization of keywords.
  4. Line 47, when EPA and DHA are first mentioned, the full name should be used. Line 63, additional abbreviation at Schizochytrium. Please standardize the writing format of the full manuscript.
  5. Please add the reasons for choosing rainbow trout as experimental animals in the introduction.
  6. Please add a description of the beneficial effects on aquatic animals of Schizochytrium dietary supplementation in the introduction. Are there any adverse effects?
  7. Are rainbow trout raised at 1 day old? I suggest that the description of the species, age, source and rearing environment of experimental animals should be listed separately in Materials and Methods.
  8. How is the feeding time of the experimental animals determined?
  9. At Line 136, Line 140, Line 142, and Line 160, there is a space between the number and the unit.
  10. Line 144, the "20 hours" format is the same as the previous one.
  11. Please standardize the correct writing of the full manuscript, such as Line 26, Line 162 and Line 163, "ml" is changed to "mL".
  12. Please supplement the sample disposal method of SOD kit in the Materials and Methods.
  13. Please add units in the table to help readers understand better.
  14. Line 201, change "table 4" to "Table 4". Same problem with Line 276 and Line 277.
  15. Please explain why the mortality of rainbow trout in the control group is so much higher than that in the experimental group.
  16. In the discussion, there is too little discussion on the results of this experiment, and many previous literatures are cited, so that the reader cannot see the focus.
  17. Please explain the reason for detecting SOD and why it is classified as non-specific immunity?
  18. It is mentioned in the conclusion that 20~80% of fish oil can be replaced by microalgae. Why is there no specific replacement rate given? The abstract prefers the beneficial effect of T20.
  19. Please improve the format of the references, such as the lack of page numbers in the third article [3].

Author Response

Dear Reviewers and Editor,

We sincerely thank you for your efforts in improving the quality of this manuscript through your thoughtful and considerate comments and suggestions. We carefully considered your comments one-by-one, performed the necessary changes in the manuscript (track-changes on) and responded to each comment here (bold font). If there are further concerns or suggestions, please don’t hesitate to let us know.

Sincerely yours,

Authors

Reviewer3

The paper "Partial substitution of fish oil with microalgae (Schizochytrium sp.) can improve growth performance, nonspecific immunity and disease resistance in rainbow trout, Oncorhynchus mykiss" was designed to evaluate whether dietary microalgae could be used as a substitute for fish oil in rainbow trout rearing. It was found that dietary microalgae had good digestibility and could replace up to 80% of fish oil without affecting the growth performance, whole-body nutritional integrity and health of the carnivorous fish rainbow trout.

1.The abstract is roughly divided into background, purpose and significance, methods, results and conclusions. The results of this abstract are too descriptive, the description of the research background is lacking, and the statement of the conclusions is not clear enough.

Response: The abstract part has been modified. Due to the limitation of the 200 words for this journal we could not add a background sentence.

2. Why are digestion experiments performed separately from other experiments?

Response: First, the goal of each experiment is different. Second, digestibility feed formulations contain a marker (chromium oxide) as mentioned in line 76. Third, to perform accurate digestibility trails with minimal stress, continuous handling of fish is required for feces stripping and juveniles can’t tolerate this stress and often get diseased and die. This will negatively influence the results of the digestibility trails. So, in digestibility trails, often larger fish is used because they are stronger and each time more amount of feces can be withdrawn. On the other hand, in growth trials, juvenile fish in more recommended because of their fast growth.    

3. Please unify the capitalization of keywords.

Response: This was modified accordingly.

4. Line 47, when EPA and DHA are first mentioned, the full name should be used. Line 63, additional abbreviation at Schizochytrium. Please standardize the writing format of the full manuscript.

Response: This was modified accordingly.

5. Please add the reasons for choosing rainbow trout as experimental animals in the introduction.

Response: According to the reviewer’s suggestion, a paragraph has been added.

6. Please add a description of the beneficial effects on aquatic animals of Schizochytrium dietary supplementation in the introduction. Are there any adverse effects?

Response: Most of the studies on SC are related to replacement of fish oil. In lines 60-66 we have mentioned several studies that have used this ingredient in fish diet. Also, we added a sentence emphasizing the rich DHA of SC.

7. Are rainbow trout raised at 1 day old? I suggest that the description of the species, age, source and rearing environment of experimental animals should be listed separately in Materials and Methods.

Response: In the Materal and Method section lines 144-156 we have mentioned the following information: size of species, source of species, type of original farm, rearing environment, conditions and etc.

8. How is the feeding time of the experimental animals determined?

Response: This was adjusted based on previous studies on rainbow trout: DOI: 10.1111/anu.12658

9. At Line 136, Line 140, Line 142, and Line 160, there is a space between the number and the unit.

Response: This was corrected according to the journal format throughout the manuscript.

10. Line 144, the "20 hours" format is the same as the previous one.

Response: Thanks for the attention to detail. This was corrected.

11. Please standardize the correct writing of the full manuscript, such as Line 26, Line 162 and Line 163, "ml" is changed to "mL".

Response: This was corrected throughout the manuscript.

12. Please supplement the sample disposal method of SOD kit in the Materials and Methods.

Response: According to the reviewer’s suggestion, two new sentences were added for this (line188-191)

13. Please add units in the table to help readers understand better.

Response: Some tables were modified. For other tables, the units for each table are presented in the header.

14. Line 201, change "table 4" to "Table 4". Same problem with Line 276 and Line 277.

Response: This was corrected.

15. Please explain why the mortality of rainbow trout in the control group is so much higher than that in the experimental group.

Response: Explanations have been added line 393-397.

16. In the discussion, there is too little discussion on the results of this experiment, and many previous literatures are cited, so that the reader cannot see the focus.

Response: Some parts of the discussion have been modified accordingly.

17. Please explain the reason for detecting SOD and why it is classified as non-specific immunity?

Response: Line 376-378: Super oxide dismutase (SOD), an antioxidant enzyme well known to control the reactive oxygen species in cells and catalyze the dismutation of hydrogen peroxide and oxygen from superoxide radicle

18. It is mentioned in the conclusion that 20~80% of fish oil can be replaced by microalgae. Why is there no specific replacement rate given? The abstract prefers the beneficial effect of T20.

Response: up to 80% of fish oil could be replaced using microalgae without any adverse effects on the fish growth performance, whole body nutrient profile and heamotological characteristics. We mentioned in the next line that 20% replacement resulted in the highest growth.

19. Please improve the format of the references, such as the lack of page numbers in the third article [3].

Response: This has been modified.

Round 2

Reviewer 1 Report

General opinion: The authors of presented manuscript did perform changes according to all comments and reasonable answered for my queries. Due to that, I add additional comments to end review.

Abstract L 13: It is always very helpful to add to abstract 1-2 open sentences about the background of conducted study. Like, what was the main reason you decide to perform such experiments?

Introduction L 30-32: Reference is needed.

Introduction L 37: no space between oil and reference

Introduction L 40: no space before reference.

Introduction L 40: due to the sentence “(..) that limits their potential as fish oil alternatives” why such characteristic is a limitation? Please provide explanation how it can affect fish health/production parameters

Introduction L 49-55: Please think about rearrange of sentences to avoid repetition of words, especially “microalgae”

Introduction L 61: Lesser attention? Maybe lower/minor could sounds better

Results Table 4: FCR should be provided onto the table

Results L 214: PER is mentioned in results description while it is not included in the table 4.

Discussion L 277: Something goes wrong with the references. It should be [23-25], not listed one by one. Please check all references in text

Author Response

Dear Reviewer and Editor,

We emphasize that fact that the quality of this manuscript has been greatly improved after your comments and suggestions. The few remaining comments by the first reviewer were addressed one-by-one and necessary changes have been done on the manuscript. We appreciate your patience and time in reviewing our manuscript.

Best regards,

Authors

General opinion: The authors of presented manuscript did perform changes according to all comments and reasonable answered for my queries. Due to that, I add additional comments to end review.

Abstract L 13: It is always very helpful to add to abstract 1-2 open sentences about the background of conducted study. Like, what was the main reason you decide to perform such experiments?

Response: As recommended by the reviewer, two lines were added to the abstract as background.

Introduction L 30-32: Reference is needed.

Response: A reference was added to this part.

Introduction L 37: no space between oil and reference

Response: We have corrected this accordingly.

Introduction L 40: no space before reference.

Response: Corrected accordingly.

Introduction L 40: due to the sentence “(..) that limits their potential as fish oil alternatives” why such characteristic is a limitation? Please provide explanation how it can affect fish health/production parameters

Response: Based on reviewer’s suggestion, we have added a sentence to explain this.

Introduction L 49-55: Please think about rearrange of sentences to avoid repetition of words, especially “microalgae”

Response: Thanks for the attention to detail; we have changed the wording of this part.

Introduction L 61: Lesser attention? Maybe lower/minor could sounds better

Response: This has been changed to minor attention.

Results Table 4: FCR should be provided onto the table

Response: We have added the FCR according to the reviewer’s suggestion.

Results L 214: PER is mentioned in results description while it is not included in the table 4.

Response: We have added the PER according to the reviewer’s suggestion.

Discussion L 277: Something goes wrong with the references. It should be [23-25], not listed one by one. Please check all references in text

Response: We have modified this accordingly and have checked the whole manuscript.

Reviewer 2 Report

Dear Authors,

thank you for your care and attention to my previous comments. I found the manuscript improved now, and your answers affordable.

I have no other suggestions for your manuscript.

Good luck!

The Reviewer

Author Response

Dear Reviewer and Editor,

We emphasize that fact that the quality of this manuscript has been greatly improved after your comments and suggestions. The few remaining comments by the first reviewer were addressed one-by-one and necessary changes have been done on the manuscript. We appreciate your patience and time in reviewing our manuscript.

Best regards,

Authors

Reviewer 3 Report

The article is modified reasonably, the results are reasonable, and the writing process is reasonable, so it is recommended that this journal accept the article.

Author Response

(The authors gave the same response as above.)
